# A Deep Bayesian Policy Reuse Approach Against Non-Stationary Agents

**Yan Zheng**[1], **Zhaopeng Meng**[1], **Jianye Hao**[1][*], **Zongzhang Zhang**[2],
**Tianpei Yang**[1], **Changjie Fan**[3]
[1]College of Intelligence and Computing, Tianjin University, Tianjin, China
[2]School of Computer Science and Technology, Soochow University, Suzhou, China
[3]NetEase Fuxi Lab, NetEase, Inc., Hangzhou, China
{yanzheng, mengzp, jianye.hao}@tju.edu.cn, zzzhang@suda.edu.cn
tpyang@tju.edu.cn, fanchangjie@netease.com

## Abstract

In multiagent domains, coping with non-stationary agents that change behaviors from time to time is a challenging problem, where an agent is usually required to be able to quickly detect the other agent's policy during online interaction, and then adapt its own policy accordingly. This paper studies efficient policy detecting and reusing techniques when playing against non-stationary agents in Markov games. We propose a new deep BPR+ algorithm by extending the recent BPR+ algorithm with a neural network as the value-function approximator. To detect policy accurately, we propose the *rectified belief model* taking advantage of the *opponent model* to infer the other agent's policy from reward signals and its behaviors. Instead of directly storing individual policies as BPR+, we introduce *distilled policy network* that serves as the policy library in BPR+, using policy distillation to achieve efficient online policy learning and reuse. Deep BPR+ inherits all the advantages of BPR+ and empirically shows better performance in terms of detection accuracy, cumulative rewards and speed of convergence compared to existing algorithms in complex Markov games with raw visual inputs.

## 1 Introduction

With recent advance of deep reinforcement learning (DRL) techniques [17, 18, 22, 24, 25], a large number of DRL algorithms have been successfully applied to solve challenging problems such as game playing [18], robotics [14] and recommendation [26]. Yet, most of these algorithms focus on single-agent domains, without explicitly considering the coexisting agents in the environments.

There also exist many application scenarios involving multiagent interactions, commonly known as multiagent systems (MAS). A few multiagent DRL algorithms [5, 19, 23] have been proposed that focus on directly searching an optimal policy, without explicitly considering coexisting agent's behaviors. However, in MAS, it is critically essential for agents to learn to cope with each other by taking the other agent's behaviors into account [2, 6, 13, 15, 16]. But none of the these works explicitly categorizes the other agent's policy. Without such an explication, a learned policy cannot be directly exploited to achieve higher cumulative reward, a.k.a., return.

In this work, we study the problem of playing against non-stationary agents, the second most sophisticated problem categorized in [9], by explicitly identifying their categorized policies, and then reusing learned strategies against them. When an agent uses an unknown policy, the optimal policy against it can be efficiently learned using a starting policy built with a *distilled policy network*. The

---

[*]Corresponding author: Jianye Hao.

BPR+ algorithm [7, 10] has similar ideas but with the following limitations: 1) BPR+'s internal belief model is updated only using the reward signal, which may be insufficient to detect the other agent's policy accurately; 2) BPR+ learns an optimal policy from scratch whenever the other agent is detected using an unknown policy, which can be inefficient especially in large domains. Additionally, it uses a model-based tabular algorithm, R-max, restricting itself to small domains; and 3) BPR+ is a tabular-based algorithm that directly stores learned policies as Q-tables, which might be space-inefficient and even infeasible when handling larger problems or storing numerous policies.

To address above limitations in BPR+, we propose a new algorithm, named deep BPR+, combining BPR+ with recent DRL techniques. To predict the other agent's policy accurately, we rectify the *belief model* in BPR+ using an *opponent model*, which encodes an agent's policy, analyzing from both aspects of the reward signal and coexisting agent's behavior to improve its detection accuracy. As for storing and reusing policies, we introduce the distilled policy network using policy distillation [21], which plays a role of the policy library in BPR+, providing convenient policy switching. Most importantly, compared with learning from scratch, it allows us to more efficiently learn new policies that can compete well against previously unseen policies. Another side benefit it brings is high spatial utilization, which makes it suitable for applications with limited storage spaces. Empirical results show that, compared to existing algorithms, deep BPR+ can achieve more accurate policy detection, more efficient policy reuse and higher speed of convergence towards optimal policies in face of a non-stationary agent using a new unseen policy in complex Markov games with raw visual inputs.

## 2 Preliminaries

**Bayesian policy reuse (BPR)** [20] provides an efficient framework for an agent to act by selecting the best response strategy when facing with an unknown task. Formally, a task $\tau \in \mathcal{T}$ is defined as an MDP, and a policy $\pi(s)$ is a function that outputs an appropriate action $a$ given state $s$. The return, or utility, generated from interacting with the MDP environment by following $\pi$ over an episode of $k$ steps, is defined as the cumulative reward $u = \sum_{i=1}^{k} r_i$, where $r_i$ is the immediate reward received by acting at step $i - 1$. BPR uses a performance model $P(U|\tau, \pi)$, which is a probability distribution over the utility $U$, to describe how policy $\pi$ behaves on task $\tau$. For a set of previously-solved tasks $\mathcal{T}$, the belief $\beta(\tau)$ is a probability distribution over $\mathcal{T}$ that measures to what extent that the currently faced task $\tau^*$ matches the known task $\tau$ based on reward signal (i.e., the cumulative reward $u$). The belief model $\beta^0(\tau)$ is initialized with a prior probability and updated at time $t$ using Bayes' rule:

$$\beta^t(\tau) = \frac{1}{\eta} P(u^t|\tau, \pi^t)\beta^{t-1}(\tau), \tag{1}$$

where $\eta$ is the normalization factor $\sum_{\tau' \in \mathcal{T}} P(u^t|\tau', \pi^t)\beta^{t-1}(\tau')$. Based on the belief model, to maximize the utility, BPR uses a policy-selection indicator, named probability of expected improvement, to reuse the most appropriate policy in a policy library $\Pi$ [20]. The indicator considers the expected hypothesized increment over the utility that the current best policy can achieve. Assume that $\bar{U}$ is the expected utility of the best current policy under the current belief measured by $\max_{\pi \in \Pi} \sum_{\tau \in \mathcal{T}} \beta(\tau)\mathbb{E}[U|\tau, \pi]$. Thus BPR chooses the best potential policy $\pi^*$, i.e., the policy most likely to result in any improvement to the expected utility:

$$\pi^* = \arg\max_{\pi \in \Pi} \int_{\bar{U}}^{+\infty} \sum_{\tau \in \mathcal{T}} \beta(\tau) P(U|\tau, \pi) dU. \tag{2}$$

BPR+ extends BPR to handle non-stationary opponents in multiagent settings and learns new performance models in an online manner. Note that the tasks and policies that BPR+ faces correspond to opponent strategies and optimal policies against those strategies, respectively. Despite BPR+ has the ability to detect the strategy switch to adjust current strategy online, its effectiveness is only tested with a tabular representation in a single-state iterated matrix game.

**Policy distillation** [21] can be used to successfully consolidate multiple task-specific policies into a single policy network by transferring knowledge, i.e., Q-function, from a teacher model $\psi$ to a student model $\phi$. The teacher $\psi$ generates a dataset $\mathcal{D}^\psi = \{(s_i, \mathbf{q}_i)\}_{i=0}^{N}$, where each sample consists of a state $s_i$ and a vector $\mathbf{q}_i$ of unnormalized Q-values with one value per action. The regression is used to train a student $\phi$ with samples $(s, a, r, s')$ drawn from $\mathcal{D}^\psi$ to produce the output similar to the teacher $\psi$. Similar to [11], the Kullback-Leibler (KL) divergence with temperature $t$ is adopted to

**Algorithm 1:** Deep BPR+

---

**Input:** Episodes $K$, policy library $\Pi$, known opponent policy set $\mathcal{T}$, performance model $P(U|\mathcal{T}, \Pi)$

1   Initialize beliefs $\bar{\beta}^0$ with uniform distribution
2   **for** *episode t = 1 ... K* **do**
3     **if** *execute a reuse stage* **then**
4        Choose a policy $\pi^t$ based on $\bar{\beta}^{t-1}$ to execute, and receive utility $u^t$ (see Equation 11)
5        Estimate opponent's online policy $\hat{\tau}_o^t$ based on its observed behaviors
6        Update rectified belief model $\bar{\beta}^t$ using $u^t$ and $\hat{\tau}_o^t$ (see Equation 10)
7        **if** *a new policy is detected by moving averaged reward* **then**
8           Initialize policy $\pi^t$ by distilled policy network, and then switch to learning stage
9     **else if** *execute a learning stage* **then**
10        Optimize $\pi^t$ by DQN, and estimate opponent's online policy $\hat{\tau}_o^t$
11        **if** *an optimal policy is obtained* **then**
12           Update $\mathcal{T}$, $\Pi$ and $P(U|\mathcal{T}, \Pi)$, and then switch to the reuse stage

---

measure the training loss between the models $\psi$ and $\phi$ using the following form:

$$Loss_{KL}(D^\psi, \theta_\phi) = \sum_{i=1}^{|D^\psi|} \text{softmax}(\frac{\mathbf{q}_i^\psi}{t}) \ln \frac{\text{softmax}(\frac{\mathbf{q}_i^\psi}{t})}{\text{softmax}(\mathbf{q}_i^\phi)}. \tag{3}$$

The $i^{th}$ element of softmax($\mathbf{z}$) is defined as $\exp \mathbf{z}(i) / \sum_j \exp \mathbf{z}(j)$, where $\mathbf{z}$ represents a vector.

## 3   Deep BPR+

Deep BPR+ extends BPR+ from tabular representation to deep neural network approximation, and addresses the following two major drawbacks in such an extension. To be consistent in the below, we use "opponent" to indicate the coexisting agent regardless of cooperative or competitive environments.

First, the accuracy of the belief model $\beta(\tau)$ in Equation 1 is highly dependent on the performance model $P(U|\tau, \pi)$, which evaluates the policy $\pi$ behaving against the opponent using policy $\tau$, named response policy. However, the performance model of a response policy against different strategies might be the same in multiagent domains, resulting in indistinguishability of the belief model and thus inaccurate detection. To address this, we detect the opponent's policy simultaneously using the belief model based on reward signals and opponent models based on observations.

Second, when learning a response policy (e.g. DQN) against a new opponent, BPR+ learns from scratch, resulting in inefficiency due to expensive training time. Here, we propose the distilled policy network (DPN), using policy distillation to combine multiple response policies into a single one. When learning a new response policy, the distilled policy network can be used to initialize a starting policy to speed up the learning process, thus significantly improve the quick response ability.

The overall flow of deep BPR+ is outlined in Algorithm 1. It consists of two stages: 1) a reuse stage (lines 3-8) selecting the most appropriate policy to execute; and 2) a learning stage (lines 9-12) obtaining an optimal response policy against a new opponent. In each episode, only one stage will be executed. During the reuse stage, deep BPR+ selects a response policy $\pi^t$ using the new belief model $\bar{\beta}^t$ rectified by the opponent model, receives cumulative reward $u^t$, estimates the opponent's online policy $\hat{\tau}_o^t$, and updates $\bar{\beta}^t$ using $u^t$ and $\hat{\tau}_o^t$. At the end of the stage, deep BPR+ checks whether the opponent is using a previously unseen policy. If yes, it switches to the learning stage in the next episode and starts the policy optimization using a starting policy $\pi^t$ initialized by DPN. In the learning stage, any DRL algorithm can be used to learn a response policy online. Once it is finished, the estimated opponent's policy $\hat{\tau}_o^t$, that depicts the opponent's new policy, will be added into the known opponent policy set $\mathcal{T}$, the corresponding learned response policy will be added into the policy library $\Pi$, and the performance model $P(U|\mathcal{T}, \Pi)$ will be updated using the corresponding received cumulative reward. At last, deep BPR+ will switch back to the reuse stage in the next episode.

Note that, deep BPR+ currently changes its policy only at the beginning of each episode since the reward signal for detection is on the episode basis. However, the policy update can also be performed at each step if the instantaneous reward is used as the signal in the belief model (line 4). The choice of the signal is problem-specific, and it determines the granularity of policy selection frequency.

## 3.1 Rectified Belief Model (RBM)

Detecting the opponent's policy is a critical part in deep BPR+, since higher detection accuracy enables more efficient policy reuse, resulting in better performance. However, in vanilla BPR+, the belief model, originally designed for measuring the similarity between different tasks in transfer learning, suffers from inaccurate detection in multiagent domains. More specifically, $\beta^k(\tau) \equiv \beta^k(\tau|u^k, \pi^k)$ in Equation 1 describes the probability of the opponent using policy $\tau$ at episode $k$ given that the agent uses policy $\pi$ against $\tau$ and receives reward $u^k$. At the beginning of episode $k + 1$, the agent chooses the most appropriate response policy to execute by reasoning about the opponent's policy $\tau^*$:

$$\tau^* = \arg\max_\tau \beta^k(\tau), \tag{4}$$

where $\tau^*$ is a unique solution only when $\beta(\tau^*) > \beta(\tau_i)$ for every $\tau_i \neq \tau^*$. However, this condition does not always hold because the belief model is updated only using the performance model in Equation 1, making it being proportional to and highly dependent on the performance model:

$$\beta^k(\tau) \equiv \beta^k(\tau|u^k, \pi^k) \propto P(u^k|\tau, \pi^k). \tag{5}$$

Assuming in a fully cooperative environment, an agent uses $\pi_1, ..., \pi_n$ policies to cooperate with opponent using $\tau_1, ..., \tau_n$ policies respectively. In this case, any miscoordination results in zero reward meaning $P(0|\tau_i, \pi_j) \simeq 1$ where $i \neq j$ and $i, j \in [1, n]$. Suppose, at episode $k$, an agent uses policy $\pi_j$ against its opponent using $\tau_i$ and thus results in a miscoordination. Since the performance model for each miscoordinated policy pair $(\tau_i, \pi_j)$ is indistinguishable as follows:

$$P(u = 0|\tau_1, \pi_i^k) = \cdots = P(u = 0|\tau_{i-1}, \pi_i^k) = P(u = 0|\tau_{i+1}, \pi_i^k) = \cdots = P(u = 0|\tau_n, \pi_i^k) \simeq 1. \tag{6}$$

This leads to the belief models over different $\tau_j(j \neq i)$ are indistinguishable as well:

$$\beta^k(\tau_1) = \cdots = \beta^k(\tau_{i-1}) = \beta^k(\tau_{i+1}) = \cdots = \beta^k(\tau_n). \tag{7}$$

In Equation 4, there exist multiple $\tau^*$ solutions. One of them will be randomly selected and thus leads to continuous miscoordination in the following episodes. So, detecting from only one single perspective of policy's performance is not enough to exactly reason about the opponent's policy.

To overcome this, we propose the opponent model/policy $\hat{\tau}$ parameterized by $\theta$, a neural network approximator to an opponent's true policy $\tau$, depicting its behaviors from its past sequence of moves. Opponent model is critical in identifying the other agent's type by estimating its policy [1]. Similar ideas can be found in recent multiagent DRL algorithms (e.g., DRON [6] and MADDPG [16]). However, they either use handcrafted behavioral features or observed agents' behaviors to learn a generalized policy without explicitly opponent classification. DPIQN [12] learns features for different opponent's policies, but uses them to train a generalized Q-network to execute, rather than directly reusing a more advantageous response policy. LOLA [4] optimizes the policy by considering that the coexisting agent learns at the same time. It belongs to the category of "theory of mind" defined by [9] which is computationally expensive. In deep BPR+, we explicitly categorize opponent's policy and try to achieve better performance by reusing the best response policy. Assuming that $(s_0, a_0, ..., s_t, a_t...)$ is the observation during interactions with an opponent using policy $\tau$, the opponent model $\hat{\tau}$ can be obtained by maximizing the log probability of policy $\hat{\tau}$. However, only using sampled observation may easily incur the over-fitting issue. Besides, observations may vary greatly among different episodes, resulting in high variance. To alleviate this, an entropy regularizer is introduced into the loss function:

$$\mathcal{L}(\theta) = -\mathbb{E}_{s_i, a_i} \left[ \log \hat{\tau}(a_i|s_i) + H(\hat{\tau}) \right], \tag{8}$$

where $H$ is the entropy of policy $\hat{\tau}$ and $(s_i, a_i)$ are training samples from the observations. With the opponent model $\hat{\tau}$, the similarity between opponent's different policies $\tau_1$ and $\tau_2$ can be measured using a KL divergence, denoted by $D_{KL}(\hat{\tau}_1, \hat{\tau}_2) \approx \mathbb{E}_{(s,a)} \log \left\{ \frac{\hat{\tau}_1(s,a)}{\hat{\tau}_2(s,a)} \right\}$, whereby opponent's different policies with same belief can be further distinguished by utilizing the corresponding opponent model. Specifically, during online interaction, line 5 in Algorithm 1 estimates the opponent's online policy

$\hat{\tau}_o$, thus the posterior probability (unnormalized) of the opponent using policy $\tau$ can be defined as follows:

$$p(\tau) \equiv p(\hat{\tau}|\hat{\tau}_o) = \sum\nolimits_{\hat{\tau}_i \in \mathcal{T}} D_{KL}(\hat{\tau}_o, \hat{\tau}_i)/D_{KL}(\hat{\tau}_o, \hat{\tau}). \tag{9}$$

Here, $\hat{\tau}_i$ is the previously learned approximate policy of the known opponent's policy $\tau_i$ stored in $\mathcal{T}$ and the KL divergence is measured using observed state-action pairs. Note that, due to the similarity between opponent's different policies being inversely proportional to the value of the KL divergence, here we thus use the reciprocal of the relative proportion of the KL divergence. Besides, the posterior probability is normalized using $\eta$ in Equation 10.

Intuitively, both belief and opponent models can be understood as the posterior probabilities measuring the opponent's policy, based on received reward signal $u$ and observed online behaviors $\hat{\tau}$ respectively. $\beta(\tau)$ and $p(\tau)$ are independent of each other due to their dependence on $u$ and $\hat{\tau}$ separately. Thus, we propose a rectified belief model that measures the probability of the opponent using policy $\tau$ from both models by directly multiplying them together to obtain a more accurate prediction model:

$$\bar{\beta}(\tau) = \frac{1}{\eta} p(\tau) P(u^k|\tau, \pi^k) \beta^{k-1}(\tau), \tag{10}$$

where $\eta = \sum_{\tau' \in \mathcal{T}} p(\tau') P(u^k|\tau', \pi^k) \beta^{k-1}(\tau')$. RBM rectifies the performance model using the opponent model to select a more appropriate policy $\pi^*$ in terms of cumulative reward to execute:

$$\pi^* = \arg\max_{\pi \in \Pi} \int_{\bar{U}}^{+\infty} \sum\nolimits_{\tau \in \mathcal{T}} \bar{\beta}(\tau) P(U|\tau, \pi) dU. \tag{11}$$

Finally, deep BPR+ uses the same moving average reward as the BPR+ to detect whether encountering an unknown policy (line 8 in Algorithm 1). Specifically, let $r_\pi^t$ be the accumulated rewards in episode $t$ using policy $\pi$. Then, the possibility of obtaining $r_\pi^t$ given the non-stationary agent's policy with label $\tau$ is $p_t^\tau = P(r_\pi^t|\tau, \pi)$. If the value of $\sum_{i=t-n+1,\dots,t} p_t^\tau/n$ (for all $\tau \in \mathcal{T}$) over the last $n$ rounds is lower than a threshold $\mathcal{P}_{thr}$, then the deep BPR+ agent infers that the other agent is using a previously unseen strategy, and switches to the learning stage to learn an optimal policy against it quickly. Intuitively, $\mathcal{P}_{thr}$ is the lower bound of the possibility of the opponent using a known policy in $\mathcal{T}$, and reflects the sensitivity of a deep BPR+ agent responding to a potentially unknown opponent.

## 3.2 Distilled Policy Network (DPN)

Deep BPR+ inherits the ability of online learning against a new opponent. However vanilla BPR+ learns from scratch each time when encountering a new opponent, which can be extremely inefficient. There may exist certain similarities among opponent's different policies, as well as their associated response policies. Thus it is reasonable to reuse the learned response policy as a starting policy for the subsequent policy optimization when its associate opponent model shares high similarity to the new opponent model. A straightforward way to achieve this, given the opponent model $\hat{\tau}_o$, is to directly reuse the policy $\pi_i$, whose associated opponent model $\hat{\tau}_i$ has the highest $D_{KL}(\hat{\tau}_o, \hat{\tau}_i)$, as the starting policy. Optimizing from a learned policy improves the learning efficiency, however, may incur insufficient explorations as well, thus resulting in suboptimal solutions. To address this issue, we introduce DPN to achieve efficient learning while avoiding being stuck in suboptimal solutions.

DPN, as depicted in Figure 1(a), is comprised of a shared convolution layer and multiple separate controller layers, each of which is trained for a distinct response policy. A different label is fed into DPN to switch among response policies by linking the shared convolution layer with the corresponding control layer, achieving convenient fast policy switch. Specifically, the response policies against opponent's $n$ different policies are trained separately and distinguished by different labels. The training samples (i.e., Q-values) generated by different response policies are assigned with corresponding labels. To consolidate multiple policies into a single DPN, supervised learning is performed to minimize the distillation loss using training data with $n$ kinds of labels simultaneously.

Intuitively, compared to training an independent convolution layer for each response policy, this architecture forces the shared convolution layer to learn more generalized features against different opponents, better describing the environment. As depicted in Figure 1(b), when encountering a new opponent, the starting policy is initialized by connecting the learned shared convolution layer in DPN with a randomly initialized control layer. By optimizing from the starting policy, deep BPR+ learns more efficiently and robustly against different opponents than directly reusing the response policy.

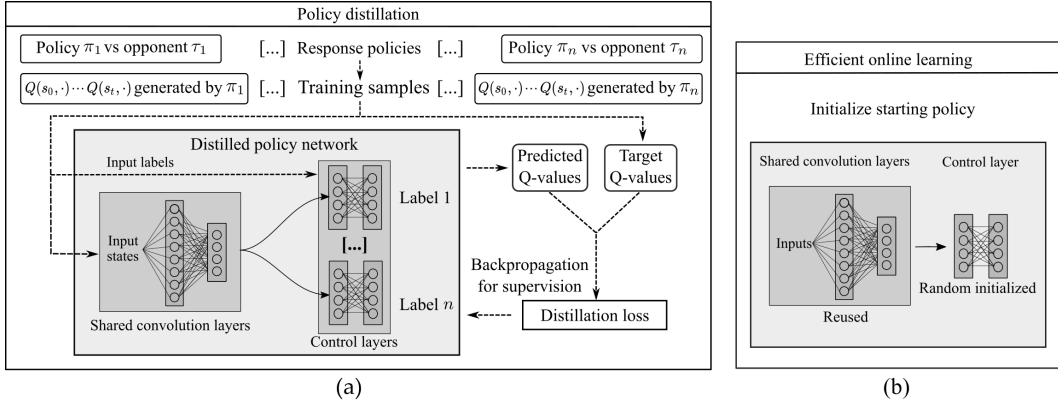

Figure 1: (a) the policy distillation in deep BPR+ and (b) the starting policy initialized by DPN.

Once the response policy is obtained, the policy distillation will be performed again to update DPN. Note that deep BPR+ does not require storing training data during online learning, because the training data can be regenerated using DPN whenever a new policy is required to be distilled. Another side benefit that DPN brings is the high spatial utilization due to its shared convolution layers.

It is worth noting that deep BPR+ has some differences from DRON [6], including: 1) deep BPR+ uses DPN to maintain accurate one-to-one response policies, while DRON uses an end-to-end trained response subnetwork, which cannot guarantee that each response policy is good enough against a particular type of opponent. 2) In DRON, parameter $K$ in the mixture-of-expert is fixed and thus cannot handle the case when the number of opponents is dynamically changing. In contrast, deep BPR+ is flexible to add any new "expert policy" into our policy library, leading to continuous performance improvement in the online fashion. 3) For policy switching, Deep BPR+ is more general as it requires no additional information except the opponent's past actions. However, DRON usually requires additional hand-crafted features, which is difficult to generalize across different domains.

## 4 Experiments

This section presents empirical results on a gridworld game adapted from [8], a navigation game adapted from [3], and a soccer game adapted from [6, 15]. Comparisons among BPR [20], BPR+ [10] and deep BPR+ are performed to verify their performance. For comparisons in multiagent DRL with raw image inputs, BPR and BPR+ here use the neural network as the function approximator. An omniscient agent, equipped with pre-trained optimally policies against the coexisting agent, is adopted as the baseline. In all games involving a non-stationary agent, deep BPR+ is empirically verified in terms of detection accuracy, cumulative reward and learning speed of a new response policy. Detailed network architecture of deep BPR+ and corresponding hyperparameters are described in Appendix.

### 4.1 Game Description

Figure 2(a) is a cooperative gridworld game, where two agents (A and O) are required to enter into their respective goals (G(A) and G(O)) while avoiding a collision. Each agent has five actions to choose: {north, south, east, west, stay}. Every movement leads the agent to move into a neighbor grid in the corresponding direction, except that a collision on the edge of the grid or thick wall results in no movement. Once entering a goal cell, the agent receives a positive reward of $+5$ and will stay there until the episode finishes. A reward of $0$ is received whenever entering at a non-goal state. Besides, if two agents try to enter the same cell, both of them stay and receive a negative reward of $-1$ as punishment. Only when both of the agents reach their goals, the episode ends. Figure 2(b) is a cooperative navigation game sharing similar settings except for some minor differences: 1) there is a thick wall (in gray) separating the area into two zones, and two agents are required to enter into the same goal (green cells) marked by "G". 2) Both agents receive a positive reward if they enter the same goal states together and then an episode ends. Otherwise, a small positive reward being proportional to their Euclidean distance is received due to miscoordination. Figure 2(c) depicts a

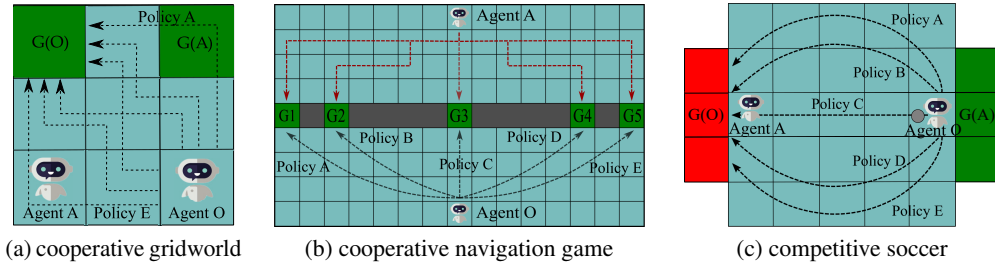

| (a) cooperative gridworld | (b) cooperative navigation game | (c) competitive soccer |

Figure 2: The configuration of tested games.

competitive $5 \times 5$ soccer game, where two agents try to steal the ball (gray circle) from each other and carry it to their respective goal areas. Different from previous settings, if both agents move to the same grid, the possession of the ball exchanges, and the move does not take place. Once an agent takes the ball to its goal, it scores a reward of $+10$ while the other agent receives $-10$ as punishment, and the game ends and resets to the configuration shown in the figure with agent O holding the ball. Intuitively, in all tested games, agent A could be able to achieve higher performance by detecting the other agent's intention more accurately.

## 4.2 Non-stationary Policy Detection

In the games shown in Figure 2, agent O has 6, 5, 5 random initialized policies (denoted in Figure 2) respectively in hand and switches its policy every several episodes. Agent A is equipped with the corresponding pre-trained response policies and aims at selecting the most appropriate policy in hand to reuse against agent O by detecting its behavior. To improve detection accuracy, deep BPR+(D) uses RBM in non-stationary policy detection and policy selection against a non-stationary agent.

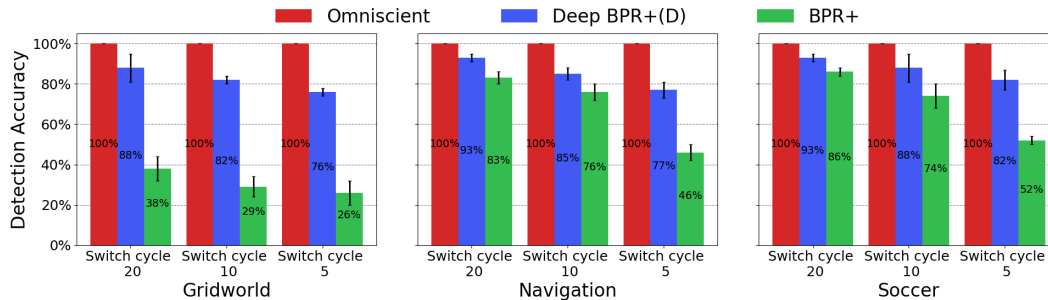

Figure 3: Comparisons of average detection accuracy (10 random seeds) against non-stationary agent.

Figure 3 demonstrates the average detection accuracy of opponent's policy, where deep BPR+(D), compared to BPR+, performs closer to the omniscient agent and achieves higher accuracy. It is worth noting that when a non-stationary agent switches its policy more frequently, the detection accuracy of BPR+ degrades dramatically while deep BPR+(D) still maintains a relatively high accuracy rate. The major reason for accuracy improvement comes from the RBM that overcomes drawbacks of vanilla BPR+, including: 1) vanilla BPR+, once failed in distinguishing opponent policy, has to try every known opponent policy, which is quite time-consuming. 2) The opponent may switch policy again before vanilla BPR+ finds the correct policy. This will perturb the update of internal belief models, resulting in further degrading of detection accuracy. Empirically, due to the RMB, deep BPR+(D) performs better by choosing the best response policy to execute, especially when the opponent switches at a fast frequency.

## 4.3 Efficient Learning against Unknown Policies

In this experiment, we evaluate the learning efficiency of deep BPR+ against agent O adopting new unknown policies during interaction. Different from the previous settings, agent A only knows, in the

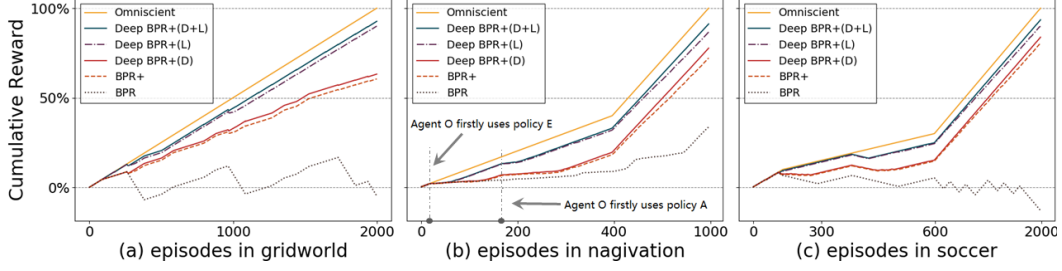

Figure 4: Comparisons of a family of BPR algorithms including BPR, BPR+, deep BPR+(D) using only the RBM in detection, deep BPR+(L) using the DPN in learning, and deep BPR+(D+L) using both. The cumulative reward is normalized by omniscient reward.

beginning, how to respond to agent O's any policy excluding policies A and E in Figure 2, and is required to learn new response policies against them in the online fashion. Following previous work [10], we assume that agent O will not switch its policy before agent A learns the new response policy. In other cases, policy switch happens after agent O executes a policy 5 to 20 times at random. The comparisons in terms of online learning speed and cumulative reward are given in Figure 4.

BPR performs poorly in the three games due to its inability to detect and respond to the agent O using unknown policies, while all the others can detect and learn the corresponding response policies against it. Note that, due to the high learning efficiency that DPN brings, deep BPR+(D+L) and deep BPR+(L) achieve higher rewards than deep BPR+(D) and BPR+. As an example, in Figure 4(b), agent O firstly uses a new policy E unknown to agent A around the $20^{th}$ episode. Deep BPR+(D+L) and BPR+(L) can detect this and efficiently learn the response policies by optimizing from the starting policy initialized by DPN, while deep BPR+(D) and BPR+ take relatively longer. The same situation happens around the $175^{th}$ episode when another new unknown policy A is adopted. Also, in the subsequent interaction, DPN can efficiently reuse the learned response policy whenever policy A or E is re-encountered. Another observation is that the RBM can achieve higher performance than the vanilla belief model regardless of whether using DPN or not.

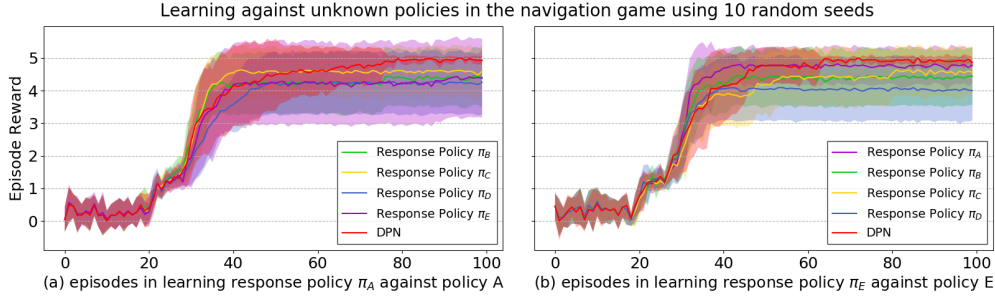

Figure 5: Comparison of learning performance when optimizing from different policies.

To further verify the benefit of DPN in learning against an unknown policy, comparisons of optimizing directly from a starting policy initialized by DPN or learned response policies are conducted and demonstrated in Figure 5. To make the comparison fair, rather than reusing the entire learned response policies, we reuse only the convolution layers in them. One observation is that, compared to optimizing directly from response policies, from a starting policy initialized by DPN achieves higher performance and stability (low variance) no matter facing unknown policy A or E.

Another observation is that, even occasionally learning from individual response policy with DQN may achieve similar performance (e.g., response policy $\pi_A$ in Figure 5(b)), any poorly selected response policy would significantly degrade the online learning performance (e.g., $\pi_B, \pi_C$ or $\pi_D$ in Figure 5). Besides, it would still be difficult to determine which response policy should be chosen as the starting policy. In contrast, DPN offers us a generalized and elegant way of initializing the starting policy, achieving promising performance without concerning about which response policy to choose.

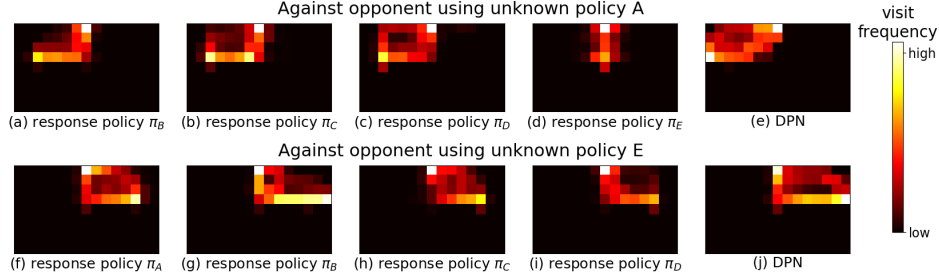

Figure 6: Visualization of trajectories when learning against unknown policies in navigation game.

In previous results, DPN shows competitive performance and robustness against different unknown policies. To investigate the intuition behind this, we visualize the trajectories of last 200 episodes when learning from different response policies $\pi_i$ as well as DPN against unknown policy A and E as shown in Figure 6. To be fair, the exploration rate during learning is the same for all settings. Intuitively, bootstrapping from learned policies trends to be stuck in local optima while DPN can avoid this and thus achieve better performance. For example, when learning against unknown policy A, response policies $\pi_B$, $\pi_C$, $\pi_D$ and $\pi_E$ seem to explore in a wrong direction, resulting in inefficient exploration (Figure 6 (a-d)). A similar phenomenon can be found in learning against policy E when optimizing from response policies $\pi_A$, $\pi_C$ and $\pi_D$ (Figure 6(f, h, i)). In contrast, DPN can achieve more efficient explorations using the same exploration rate and thus obtain better results. We hypothesize that this is because the network architecture allows the shared convolution layers to learn more generalized features against different opponents, better describing the environment and guiding the agent to explore in the right direction.

## 5 Conclusion and Future Work

This paper proposes a deep BPR+ algorithm to play against non-stationary agents in multiagent scenarios. The rectified belief model is introduced by utilizing the opponent model, achieving accurate policy detection from the perspectives of received signal and opponent's behaviors. And the distilled policy network is proposed as the policy library, achieving efficient learning against unknown policies, convenient policy reuse and efficient policy storing. Empirical evaluations demonstrate that the deep BPR+ algorithm indeed achieves promising performance than other existing algorithms on three complex Markov games. As a future work, we would like to investigate how to act optimally in face of the adaptive agents whose behaviors are continuously changing over time.

**Acknowledgments**

The work is supported by the National Natural Science Foundation of China (Grant Nos.: 61702362, 61876119, 61502323), Special Program of Artificial Intelligence, Tianjin Research Program of Application Foundation and Advanced Technology (No.: 16JCQNJC00100), Special Program of Artificial Intelligence of Tianjin Municipal Science and Technology Commission (No.: 569 17ZXRGGX00150), Science and Technology Program of Tianjin, China (Grant Nos. 15PT-CYSY00030, 16ZXHLGX00170), Natural Science Foundation of Jiangsu (No.: BK20181432), and High School Natural Foundation of Jiangsu (No.: 16KJB520041)

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
