[Supplementary Material · supplementary-appendix.pdf]

# 1 Appendix

## 1.1 Distilled policy network

Details of the DPN's architectures and the number of parameters are given in Table 1. We use three hidden convolution layers (using rectifier non-linearities between each two consecutive layers), and a fully-connected hidden layer consisting of 512 rectifier units. The output layer is a fully-connect linear layer with a single output for each valid action (5 in the benchmark).

Table 1: Network architectures and number of parameters

| # Network | Visual input | Filters in Conv. 1/2/3 | Unit in F.C. |
|---|---|---|---|
| DQN | 84 * 84 * 3 | 32/64/64 | 512 |
| Distilled Policy Network | 84 * 84 * 3 | 16/16/16 | 128 |

## 1.2 Pre-trained response policy for distillation

Table 2: hyperparameters of DQN in training response policies

| Epoch length | 2000 episodes | Discount factor | 0.99 |
|---|---|---|---|
| Optimizer | Adam | Learning rate | 0.0001 |
| $\epsilon$-greedy | 1.0(Initial) $\rightarrow$ 0.1(10000 steps) | Target network | Update every 4 steps |
| Replay memory | 50000 samples | Minibatch size | 32 |

The policy used for distillation is obtained by DQN (with $\epsilon$-greedy) with the $\epsilon$ annealed linearly from 1 to 0 over the first 200 steps. We used the Adam algorithm with 0.0001 learning rate and the mini-batches of size 32. We trained for a total of 2000 episodes and used a replay memory of 10 thousand most recent frames. At last, the softmax operation is performed when distilling with KL loss functions.

## 1.3 Online policy learning against unknown opponent.

Table 3: hyperparameters of DQN in online policy learning

| Epoch length | 500 episodes | Discount factor | 0.99 |
|---|---|---|---|
| Optimizer | Adam | Learning rate | 0.0001 |
| $\epsilon$-greedy | 1.0(Initial) $\rightarrow$ 0.1(200 steps) | Target network | Update every 4 steps |
| Replay memory | 50000 samples | Minibatch size | 32 |
| Threshold | 0.9 | | |

The online learning of the new response policy (DQN) against an unknown opponent's policy shares the same hyperparameters in Section 1.2 except that the maximum epoch length is set to 500 and the $\epsilon$-greedy drops rapidly in the first 200 steps. Besides, we assume the optimal response policy is obtained and stop the online learning if the cumulative rewards do not grow in consecutive $n$ episodes ($n = 50, 100, 200$ for gridworld, navigation and soccer games) after the initial exploration phase or the total episodes reach maximum epoch length (1000 in online learning). The threshold $\mathcal{P}_{thd}$ is set to a constant value of 0.9 to detect unknown opponent. Empirical evaluations show that 0.9 is robust to perform well across all three domains. Note that, to make the comparison fair, all algorithms in comparison use the same hyperparameters.