[Reviews · NeurIPS 2018]

Reviewer 1



The paper focuses on an important problem in multiagent learning - non-stationarity introduced by other agents. It proposes a novel rectified belief model to overcome the problem of indistinguishability with miscoordinated policies and combines a few ideas made popular by neural networks - sharing weights and distillation. This results in an extension of the idea of Bayesian Policy reuse, originally formulated for transfer learning and later extended into BPR+ for online learning, which the paper terms Deep BPR+. The paper tests the efficacy of their approach on relatively small tasks and finds that the proposed method can perform quite close to an omniscient one. The paper clearly traces the origin of its ideas to BPR and BPR+ algorithms and the limitations it's trying to overcome. It comes up with a novel approach to a belief model about other agents using both the reward signals and the opponent models based on their observations. The paper notices that in the original formulation, the belief model and the performance model are highly correlated and any miscoordination will therefore result in 0 reward and remain indistinguishable from other failures. Although it introduces another assumption which the paper should clarify - the agents have access to other agents' observations and actions (past sequence of moves) and is not trying to reason from _their_ observations of the other agent's behavior. It's slightly unclear how the thresholding is determined for switching behavior. Moreover, there are still quite a few moving parts, so hopefully the source code to replicate the experiments will be made available. The experiments, however clearly show that reasonable policies are being learned with this method and glad that the paper includes error bars in the figures. There are a few relatively minor things that are not clearly stated in the paper. For example what's the importance of entropy regularizer in opponent modeling. Or why exactly Deep BPR+ gets higher accuracy rate with faster switching. Is it just because of the distillation? Moreover it's unclear from Figure 4, how the error bars look for these methods, how many runs and seeds were used to compare. Since all experiments were done on gridworlds with only 2 agents, it's hard to say how well the proposed methods generalize to other scenarios. Also it's not clear what's meant by "coexisting agents that can adaptively change their behaviors based on their observations". I assume it's the general multi-agent learning scenario where the agent behaviors are continuously changing over time.

Reviewer 2



The paper describes a new version of BPR+, a Bayesian Policy Reuse algorithm for adapting to opponent/teammate behaviors in multiagent settings. It adds two elements: an opponent model, and a distilled policy network, that help improve some shortcomings of the previous versions. The paper is well-written, the motivation and problems with the previous algorithms are well-articulated. On a positive note, the experiments do a good job of showing the strengths of each of the new elements of Deep BPR+, in three domains, mixing both cooperative and competitve, testing each component separately. It is clear how much each element contributes in these problems. The experiments against online unknown policies is especially valuable and complementary to the motivation. As a down side, the novelty is somewhat low because the main algorithms already exist, and the new elements here are straight-forward adaptations of ideas from deep learning architectures that have been used in several other places already. After reading the start of Section 3, I was expecting that adding deep learning would mean the ability to scale to larger or more complex domains. That may be true, but it's not made clear by this paper (nor is it motivated as such). Minor comments: - "LOLA[4] studies the emergence of Nash equilibrium in iterated prisoners' dilemma". This is not quite true; it found *tit-for-tat* in IPD, which is much more surprising. It found the only mixed Nash equilibria in iterated matching pennies, but in any case this was not necessarily intended and there is no proof that LOLA is trying to find an equilibirum. Questions: - The construction of the distilled policy network sounds very similar to the architecture behind DRON. Can you clarify the main difference(s)? DRON seems to have a mixture-of-experts component used to decide how to switch.. is this also true for DPN? - Is the Omniscient policy in the experiments exact? (i.e. computed in the tabular form? If so, how was it computed (value iteration?) Suggested future work: 1. I would encourage the authors to look into more challenging domains either with partial observability or more than two players where the gains from opponent modeling are more clear. 2. On a similar note, I encourage the authors to relate or combine this approach with recent game-theoretic approaches: the overall process resembles fictitious play and/or the oracles algorithm in [1,2], where the DPN can be interpreted as the average or meta-policy (here it acts as a natural baseline to start from, but in self-play, could also provably converge to an equilibrium?). However, there was no explicit opponent modeling in those works, so Deep BPR+ can enhance these methods while potentially keeping the guarantees. [1] Heinrich & Silver 2016. Deep Reinforcement Learning from Self-Play in Imperfect-Information Games. https://arxiv.org/abs/1603.01121 [2] Lanctot et al. 2017. A Unified Game-Theoretic Approach to Multiagent Reinforcement Learning. **** Post-rebuttal: Thank you for the clarifications. I raised my score after the discussion, upon reflecting on the novelty (rectified belief update) and the wording attached to the numbers: I think this could work could be of interest to the growing multiagent learning community at NIPS.

Reviewer 3



The paper describes a new algorithm called deep BPR+ (Bayesian policy reuse). It is an extension of the earlier BPR+ algorithm that uses deep neural network. The algorithm maintains a distribution over returns for different policy-environment pairs. In each episode, it uses this distribution to compute a posterior over the possible environments and to compute a best response (policy). When facing a new environment, a new distribution over possible returns is computed as well as a new response policy. The approach is interesting, but there is one aspect that I find questionable. The paper proposes to compute a "rectified" belief over the possible environments. This belief consists of the posterior based on the observed return (which makes sense) times another term that takes into account the history of state-action pairs. This term is obtained by computing a KL-divergence. Is there a justification for this KL divergence? I can see the intuition, but this feels like a hack. Wouldn't it be possible to use Bayes' theorem to compute a proper posterior based on the history of state-action pairs? I also find confusing the presentation in terms of multi-agent systems. The paper uses the notion of opponents throughout, which suggests that there are strategic adversaries. However, the paper acknowledges that they use the term opponent also for cooperative settings, which is confusing. Perhaps even more confusing, the paper eventually confirms that the opponents have fixed policies that are changed once in a while at random. Hence, the opponents are not strategic in any way. This means that the paper does not really deal with multi-agent systems since the agents can simply be thought as environments in the sense of single agent problems. When opponents change policies at random, it simply means that the dynamics of the environment change at random. To avoid any misunderstanding, it would be better to simply write about environments whose dynamics may change instead of opponents or multi-agent systems that are non-stationary. The approach is good overall. This is incremental work that clearly improves and generalizes BPR+ with the use of deep neural networks. The proposed architecture is reasonable, though not really novel. The experiments demonstrate the effectiveness of the approach. The paper is clearly written, but there are many typos. I recommend to proofread the paper carefully and to apply a spell checker.